# Enhancing property and activity prediction and interpretation using multiple molecular graph representations with MMGX
Apakorn Kengkanna ⓘ & Masahito Ohue ⓘ ✉

Graph Neural Networks (GNNs) excel in compound property and activity prediction, but the choice of molecular graph representations significantly influences model learning and interpretation. While atom-level molecular graphs resemble natural topology, they overlook key substructures or functional groups and their interpretation partially aligns with chemical intuition. Recent research suggests alternative representations using reduced molecular graphs to integrate higher-level chemical information and leverages both representations for model. However, there is a lack of studies about applicability and impact of different molecular graphs on model learning and interpretation. Here, we introduce MMGX (Multiple Molecular Graph eXplainable discovery), investigating the effects of multiple molecular graphs, including Atom, Pharmacophore, JunctionTree, and FunctionalGroup, on model learning and interpretation with various perspectives. Our findings indicate that multiple graphs relatively improve model performance, but in varying degrees depending on datasets. Interpretation from multiple graphs in different views provides more comprehensive features and potential substructures consistent with background knowledge. These results help to understand model decisions and offer valuable insights for subsequent tasks. The concept of multiple molecular graph representations and diverse interpretation perspectives has broad applicability across tasks, architectures, and explanation techniques, enhancing model learning and interpretation for relevant applications in drug discovery.

Advanced artificial intelligence (AI) techniques have been integrated into drug discovery to facilitate various tasks, particularly the prediction of chemical properties and activities. These methods exhibit the capacity to handle multidimensional data and complex chemical space using mathematical techniques, thereby accelerate the research process. The applications of AI enable high-throughput results, cost reduction, time savings, and minimization of unintentional human errors during biochemical experiments[1–3]. Several deep learning techniques for molecular property and activity prediction have been proposing using different kinds of molecular featurizations, for instance, SMILES-based[4], fingerprint-based[5], knowledge-based[6], functional group-based[7], or image-based[8] methods. One of the potential AI techniques that has been widely used in this field is graph neural network (GNN) which encodes the compounds with molecular graph

representation[9]. GNNs helps leveraging the relationships and aggregated information between nodes and edges of the molecular graph and have demonstrated remarkable performance across numerous tasks in property and activity prediction[10,11]. When employing GNNs for compound property and activity prediction, careful consideration must be given to the way of representing molecules in graph structures as it highly influences both model learning and model interpretation.

Generally, the chemical compounds in GNNs are encoded in form of atom-level molecular graph representation by transforming atoms into nodes and bonds into edges, similar to natural form of molecules. This representation has been utilized in many research areas, such as, molecular property prediction and drug-target affinity prediction[12,13]. While the common topology of compound can be captured by this representation, it

Department of Computer Science, School of Computing, Tokyo Institute of Technology, Kanagawa 226-8501, Japan.
✉ e-mail: ohue@c.titech.ac.jp

overlooks higher-level information pertaining to chemical substructures, such as functional groups, chemical fragments, or pharmacophoric features, which are the relevant characteristics for identifying compound property and interaction[14]. This limitation can impede model from effectively recognizing information from molecular graphs. To address this, the learning layer of GNNs should be increased to encompass larger substructures. However, increasing number of layers can lead to other challenges like over-smoothing, neighbors-explosion, and over-squashing[15], while reducing number of layers may also cause under-reaching as well[16]. Besides, because of the representation at atom-level, the interpretation are sometimes scattered and inconsistent within the same functional groups or substructures which may cause confusion[17]. Therefore, many current researchers proposed alternative graph representations using reduction techniques. These representations encode original atom-level molecular graph with higher level of abstraction by simplifying subgraphs into single nodes, while preserving topological properties through predefined rules, for example, functional group, structure-based transformation, or pharmacophoric features[18,19]. Several reduced molecular graphs have been purposed, offering varying degrees of information, specificity and aggregation. Due to the coarsening of features in reduced graph, some information is discarded, and the resulted graph may be incomplete. Obviously, with different advantages and drawbacks from both graphs, choosing the approach for representing molecular structure is the essential task which affects model learning and interpretation.

For the effects on model learning, because of pros and cons from both graphs, many studies exploit the multiple molecular graph combination model by using both atom-level and another reduced graph representations in feature construction to support the model training. This approach leads to the improvement of the performance in many drug discovery tasks. For example, some studies apply the integration of pharmacophore-related graphs/features for molecular property prediction[19,20] or ligand-based virtual screening, utilizing graph edit distance (GED) as the molecular similarity calculation[21]. Other examples involve employing junction tree graphs to support the molecular generation process[22] or enhancing property prediction with a multi-level combination of message-passing neural networks[23]. Additionally, functional group-based graphs have been also integrated to improve molecular property learning[16] as well as drug-drug interaction prediction tasks[15]. Some methods develop motif-based substructure extraction creating hierarchical topology representations for property prediction[24]. However, different reduced molecular graphs may not provide suitable applicability for all tasks as they give different level of information. Furthermore, there is still a lack of research on the analysis and comparison studies of different multiple molecular graph combination models to support model learning, which limits the understanding of the effectiveness of each graph combination and selection for model implementation.

Another effect is on model interpretation. Interpretability is currently crucial aspect of GNNs when designing prediction model. Due to their inherent complexity, GNNs are often seen as black-box models with limited interpretability. This results in limitation of understanding the underlying reasons behind predictions which could be an obstacle for model refinement and acceptance in certain applications. To overcome this, interpretation techniques have been introduced to help providing rationales behind predictions and raising model transparency for explanation[25,26]. Interpretation in the context of drug discovery should provide chemically intuitive explanation telling which parts of molecule the model focuses on. These explanations should be evaluated and aligned with background knowledge. On top of that, the interpretation should give some inspirations for further related tasks. Integration of interpretation brings numerous advantages including capturing essential chemical features and scientific insights, presenting actionable guidance for the next optimization process, aiding in debugging and mitigating bias when developing model, and fostering confidence, safety, and trust[25,27]. Several publications have explored interpretation for GNNs[28,29], especially for attention mechanism techniques[27,30,31]. One of the major factors that largely determine the interpretability of GNNs

is the molecule representation. The level of molecular graph representation could be a limiting factor for model interpretation[32]. Interpretation solely on atom-level molecular graph can be sparse and inconsistent within substructures[17]. Moreover, the features of individual atoms and bonds only partially align with the understanding of chemists, leading to incomplete explanations. Thereby, integrating reduced molecular graph representations incorporating meaningful chemical features into nodes could provide better and consistent interpretation on substructures, which are informative and chemist-friendly[32]. Some studies working on multiple-graph combination models also justify their model decision-making with interpretation modules[19,30,33]. However, despite the utilization of multiple graphs in these studies, there is a dearth of research on the interpretation analysis of different molecular graph representations, leaving the influence of graph combination on model interpretation still unclear. In addition, most attention-based interpretations do not focus on cumulative explanations on the whole dataset but provide the explanation only on a specific view of single prediction which may not fully imply the overall learning of the model. Additionally, there is still less analysis regarding the statistical evaluation of interpretation on real pharmaceutical endpoint datasets, and some studies providing only interpretations without applying the results further in drug design applications. All those issues limit the comprehension and benefits of the model insights for further applications.

With aforementioned motivations and limitations, the objectives of this study are to improve understanding and representation selection of different multiple molecular graphs with GNNs for enhancing model learning and interpretation for ligand property and activity prediction tasks, and to elevate model explanations using diverse perspectives and evaluations providing more chemically meaningful and useful insights from the model that could facilitate drug discovery tasks. The overview of our proposed method, named MMGX (Multiple Molecular Graph eXplainable discovery), is depicted in Fig. 1. Three major contributions of this study consist of:

- Introducing and comparing different molecular graph representations including Atom, Pharmacophore, JunctionTree, and FunctionalGroup graph, considering information levels, features, and applications.
- Applying multiple molecular graph representations in GNNs and conducting extensive experiments on real-world ligand property/activity and synthetic datasets to validate the models, knowledge, and explanations.
- Analyzing interpretation results from attention-based mechanism with different molecular graph representations in many perspectives which are single prediction, node features and potential substructures view, evaluating them with background knowledge and ground truths, and suggesting suitable applications of interpretation to facilitate the subsequent tasks in drug discovery.

## Results and Discussion
### Datasets
When executing the study about interpretation, three main verifications should be considered, consisting of model verification for measuring model performance, knowledge verification for comparing model learning with background knowledge, and explanation verification for evaluating interpretation statistically[34]. To achieve that, there are three groups of datasets serving different goals of verification as followings.

1. **General benchmark datasets from MoleculeNet**[35]. This group of datasets is widely utilized by advanced small molecule property prediction models, so it is mainly tested for model verification. There are five benchmark datasets in different categories for this study.
2. **Pharmaceutical endpoint tasks with reported key structural patterns**. This dataset can be employed for knowledge verification because there are various publications reviewing on key structural patterns, structural alerts, or important regions in interaction maps for binding affinity which can be used as background knowledge. This group of datasets is obtained from various sources containing ten datasets.

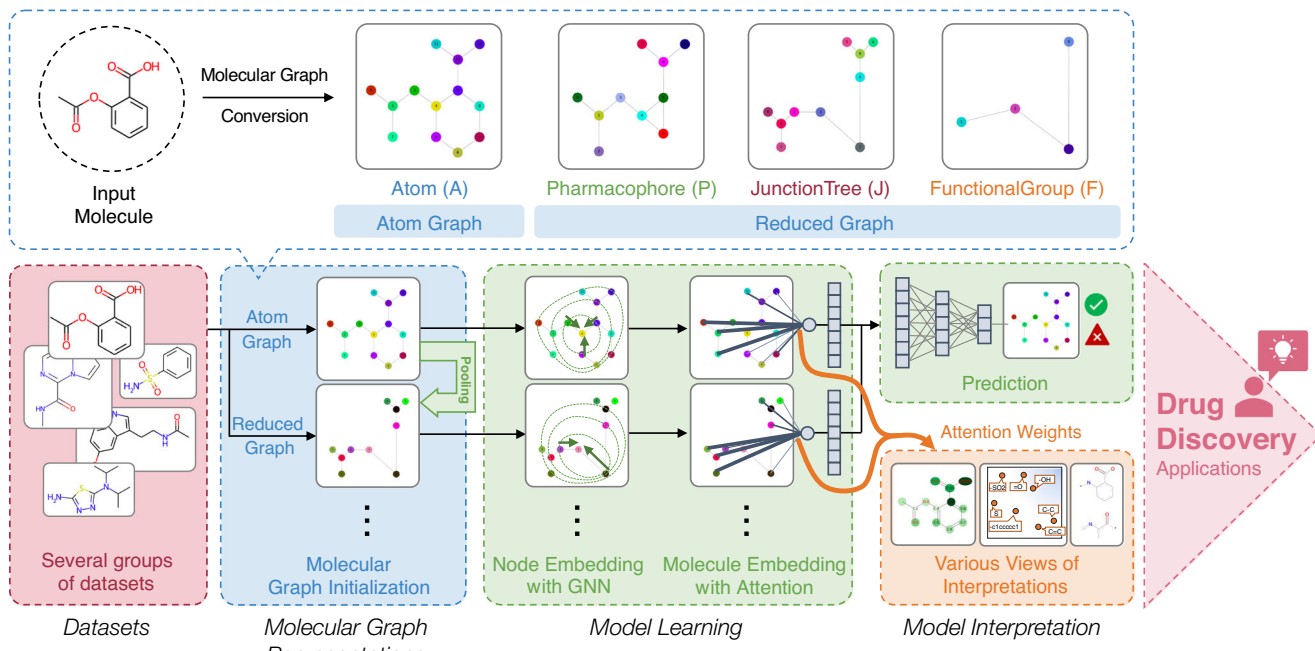

**Fig. 1 | Schematic of MMGX.** The research is conducted using several groups of datasets. Each dataset contains input small molecules which are converted into different molecular graph representations. The embedding features from graph representations are learned through graph neural networks with attention mechanism. The final molecule embedding features are used for prediction. For interpretation, attention weights from attention mechanism are extracted and visualized on different views to provide understanding of model and facilitate for subsequent tasks in drug discovery applications.

3. **Synthetic binding logics with known ground truths**. This group of datasets are constructed based on predefined logics containing five datasets, so the exact important substructures for each task are already determined. Although these datasets are simple and synthetic, it is useful for quantitative explanation verification and model understanding as it can be statistically evaluated by using known answers.

The dataset details and statistics are summarized in Table 1. Additional information regarding the description of each dataset is provided in Supplementary Table S1.

### Molecular graph representations
**Introduction to molecular graph.** Molecules can be represented in various graph topologies to describe chemical structures using relationship and adjacency between nodes and edges. Different reduction techniques can be utilized to provide abstract graph representations in which the original subgraph from the atom-level graph is contracted to be a new node that holds more interpretable higher-level information related to graph theory and chemical concept. Four distinct molecular graph representations are introduced for this study as shown in Fig. 2. The characteristics and feature description are described below with the summary shown Supplementary Table S2.

1. **Atom graph (A)** is the most general molecular representation illustrating the atoms and bonds as node and edge respectively. Node and edge features are obtained from atom and bond properties. This graph has been applied in many applications including molecular property prediction and drug-target interaction[13,36]. Atom graph maintains all topological information and substituent positions which is like the natural chemical structure. In spite of that, some limitations exist, for instance, lack of substructure information, problems with graph-based model learning and aggregation when adjusting inappropriate learning depth, and inconsistent interpretation.

2. **Pharmacophore graph (P)** is the reduced graph constructed from binding activity and pharmacophoric features using the extended

reduced graphs (ErG) algorithm[37]. The node features are embedded with one-hot encoding of six pharmacophore properties which are H-bond donor, H-bond acceptor, positive, negative, hydrophobic, and aromatic. This graph demonstrates effectiveness in scaffold hopping and protein-ligand interaction tasks[19,38]. Although this representation provides low dimensional descriptor vector with generalization and association of active compounds, the pharmacophore nodes are limited to only six types and the interpretation of this graph is challenging.

3. **JunctionTree graph (J)** is tree-based reduced graph generated by converting bonds, rings, and junction atoms into nodes, so that the final graph contains no loop structure. This kind of representation shows great performance in molecule generation and molecular property prediction tasks[22,39]. This tree-structure is beneficial in preventing the dead-loop problem and repeated information issues in message-passing process[14,30]. However, JunctionTree graph still has drawbacks in lack of larger meaningful substructures like functional groups and ring types, as well as difficulties when representing complex rings.

4. **FunctionalGroup graph (F)** is another reduced graph representation integrating functional group information. The original substructures based on predefined functional groups, ring types, and atom pairs are converted into a single node. There are several applications using FunctionalGroup graph and showing good performance including small molecule property prediction and molecular graph generation[15,40,41]. The advantage of this graph is that the node features are encoded with chemical background promoting higher-level understanding. By the way, the graph construction is limited to only predefined substructures since all chemical functional groups or complicated rings are hard to be completely predefined.

Various graph reduction and node labeling schemes can differ in terms of specificity and discrimination levels[18]. To elucidate these differences, the summary of differences between molecular graph representations is provided in Supplementary Table S3. It is evident that, despite offering

## Table 1 | Datasets details and statistics

| Group | Dataset | Category | Task | #Compounds |
|---|---|---|---|---|
| **General benchmark datasets from MoleculeNet** | BACE[35] | Biophysics | Classification | 1513 |
| | BBBP[35] | Physiology | Classification | 2050 |
| | FreeSolv[35] | Physical chemistry | Regression | 642 |
| | ESOL[35] | Physical chemistry | Regression | 1128 |
| | Lipo[35] | Physical chemistry | Regression | 4200 |
| **Pharmaceutical endpoint tasks with reported key structural patterns** | AmesMutag[61] | Physiology | Classification | 6512 |
| | hERG20[62] | Biophysics | Classification | 6548 |
| | CYP2C8[42] | Biophysics | Classification | 553 |
| | CYP3A4[63] | Biophysics | Classification | 9122 |
| | Hepatotoxicity[64] | Physiology | Classification | 1489 |
| | ROCKII[65] | Biophysics | Classification | 3953 |
| | HumanPPB[27] | Biophysics | Regression | 3921 |
| | AqSolDB[49] | Physical chemistry | Regression | 9982 |
| | HIV1[66] | Biophysics | Regression | 2602 |
| | JAK1[67] | Biophysics | Regression | 8011 |
| **Synthetic binding logics with known ground truths** | Logic6[1][57] | Synthetic | Classification | 4326 |
| | Logic7[2][57] | Synthetic | Classification | 8671 |
| | Logic9[3][57] | Synthetic | Classification | 8687 |
| | Logic14[4][57] | Synthetic | Classification | 16,598 |
| | 3MR[5][50] | Synthetic | Classification | 2877 |

[1]Logics: [FX1] and [CX3]=O.
[2]Logics: [R0;D2,D1][R0;D2][R0;D2,D1] and [CX3]=O.
[3]Logics: [NX3;H2] and [OD2](C)C and [cX3]1[cX3H][cX3H][cX3H][cX3H]1.
[4]Logics: ([OD2](C)C or no [OX2H]) and [CX3]=O and (no [CX2]#[CX2]).
[5]Logics: *1**1 (3-membered ring).

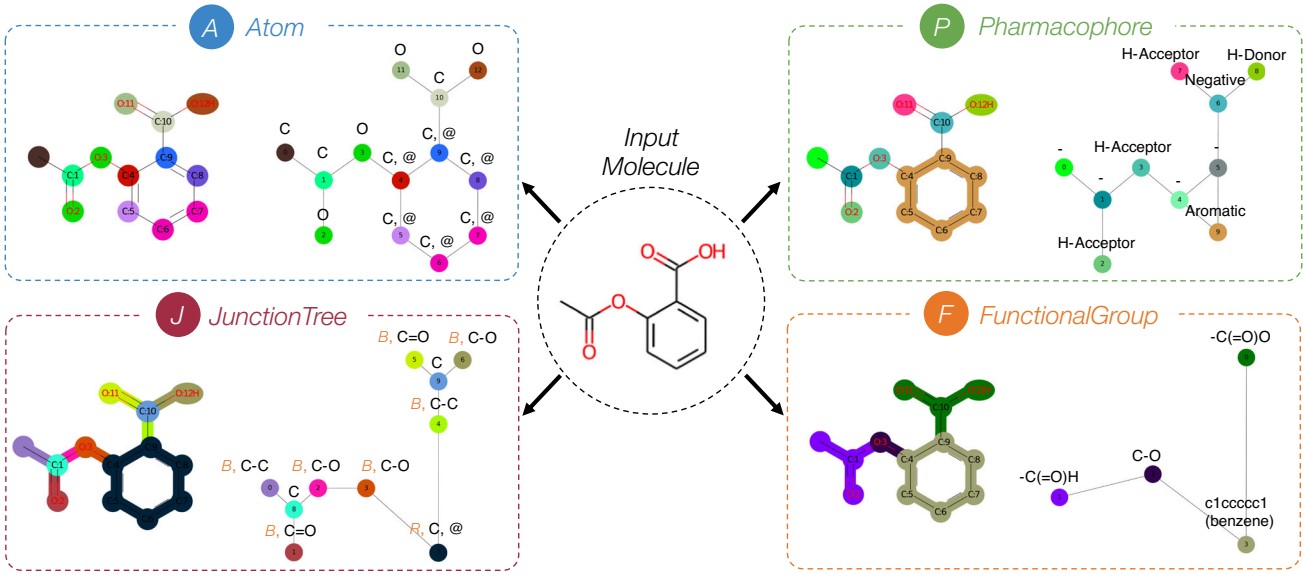

**Fig. 2 | Molecular graph representations.** Illustrations of different molecular graph representations of aspirin molecule (SMILES notation: *CC(=O) OC1=CC=CC=C1C(=O)O*) including Atom Graph (A), Pharmacophore Graph (P), JunctionTree Graph (J), and FunctionalGroup Graph (F).

complementary views, all molecular graph representations contain different sets of node and edge features, which would certainly affect model learning and interpretation.

**Molecular graph reduction analysis.** To quantify the change of coarsened graphs when applying reduction techniques, the number of nodes after reduction in each dataset are analyzed. As shown in Supplementary

Fig. S1, the average number of nodes from all molecules in different molecular graphs from general benchmark datasets are compared. The result shows that average number of nodes in reduced molecular graphs are significantly shrunk from the Atom graph. The most reduced molecular graph is FunctionalGroup graph as it obviously simplifies larger substructure of atoms into one node. On the other hand, the least reduced molecular graph is Pharmacophore graph as it mainly converts a

**Table 2 | Summary of model performance ranking of 2-graph models for physical chemistry, biophysics, physiology category datasets with MMGX**

| Model | Physical chemistry | | Biophysics | | Physiology | |
|---|---|---|---|---|---|---|
| | AvgRank | AvgZScore | AvgRank | AvgZScore | AvgRank | AvgZScore |
| A | 3.00 | −0.6793 | 3.13 | −0.4856 | 3.67 | −0.8079 |
| A+F | 2.50 | 0.1644 | 2.25 | 0.4613 | 1.33 | 1.0827 |
| A+P | 2.00 | 0.3952 | 2.00 | 0.1481 | 1.67 | 0.6556 |
| A+J | 2.50 | 0.1197 | 2.63 | −0.1238 | 3.33 | −0.9304 |

Note: The underlined numbers are the best performance of each dataset.

single atom node to a corresponding pharmacophoric feature node, except for ring system, while maintaining topological distance. This indicates that the molecular graphs are reduced by aggregating atom or node information to higher-level of abstraction.

## Model performance

This section reports the model learning performance from different experimental settings. There are four main experiments including 2-graph combination models, multiple-graph combination model, different model architectures, and comparison with other models. To present the concluded results, the average ranking (AvgRank), measuring the rank of model performance, and average z-score (AvgZScore), measuring how far of performance improvement is relatively from the mean, are employed.

**Model performance of 2-graph combination models.** To compare the effects on model learning performance of different graphs, the first experiment is conducted on four schema which are Atom graph (A), the combination of 2-graph models including Atom with FunctionalGroup graph (A +F), Atom with Pharmacophore graph (A+P), and Atom with JunctionTree graph (A+J). This research assumes that Atom graph should be the base graph as it contains fundamental information and topology, and using reduced graph alone is undesirable because some information is discarded and insufficient. To better analyze the performance of each graph representation for drug discovery task, the general benchmark and pharmaceutical endpoint tasks datasets are arranged into three categories based on MoleculeNet including physical chemistry, biophysics, and physiology as shown in Table 1. The performance is presented in Table 2 and more details in Supplementary Table S4-S6. Overall, the combination of Atom graph with another reduced molecular graph models achieve moderately better performance compared to using Atom graph alone due to the average ranking and z-score. When analyzing for particular categories, model A+F seems to outperform others in biophysics and physiology. For biophysics category, binding activity tasks would get advantages from additional information of resembling chemical structures like functional groups. For physiology category, this kind of properties, e.g., toxicity, can be partially implied from special functional group patterns, so FunctionalGroup graph would provide better benefit for these tasks. In term of physical chemistry, model A+P dominates in this category. This would be because of pharmacophoric features such as H-donor/acceptor and positive/negative features which greatly support these tasks.

To provide the overall conclusion, the average ranking and z-score performance are calculated for overall tasks as illustrated in the Table 3 on MMGX column. Upon analyzing all categories collectively, the A+P model appears to perform the best average rank 1.93, but A+F model outperforms in average z-score at 0.5064. Model A+P and A+F incorporates additional meaningful information about pharmacophoric features, functional groups, and aromatic rings, which may be challenging for the model to learn independently. Therefore, this inclusion of handcrafted-information proves beneficial for model learning.

**Model performance of multiple-graph combination models.** Next, to test the effectiveness of multiple molecular graph representations model,

the reduced graph only models (F, J, P), the combination of 3-graph models (A+F+J, A+F+P, A+J+P), and the combination of 4-graph models (A+F+P+J) experiments are conducted and tested on general benchmark datasets from MoleculeNet. The results are compared with the previous results of the Atom graph model and the combination of 2-graph models and displayed in Supplementary Table S7. As a result, while the integration of 3-graph and 4-graph models shows promising performance, they do not significantly improve the performance with this proposed model based on average ranking and z-score. Additionally, performance variations are observed across datasets. This variability may occur when the model encounters complex and irrelevant information. Therefore, it is crucial to consider appropriate feature selection and more suitable integration techniques. For this analysis, 2-graph models are considered as sufficient scheme based on these results. On the other hand, models based solely on reduced graphs exhibit poorer performance. This implies that relying just coarsened information may lead to incompleteness, which is unfavorable for these tasks.

**Model performance of different model architectures.** Even though the experiments are conducted in custom model architecture integrated with attention mechanism, the concept of combination graph model are actually applicable for any kind of graph-based models. To support this concept, further experiments of multiple molecular graph representations are conducted on simple GCN model and advanced AttentiveFP[36] model with general benchmark datasets from MoleculeNet. The model performance is displayed in Supplementary Table S8-S10. In short, integrating multiple graphs demonstrates generally good performance in all different architecture, especially benefiting simple GCN model. Thus, the concept of multiple graphs could help improve model performance positively.

To support the conclusion of the most promising and recommended combination, experiments with different model architectures may help emphasize our findings. Therefore, additional experiments of the 2-graph combination model are conducted with GCN and AttentiveFP model architectures across all datasets. The model performances for all other datasets in pharmaceutical endpoint tasks, besides general benchmarks from MoleculeNet, are recorded in Supplementary Table S11-S12. The average ranking and z-score are calculated for summarization as shown in Table 3. Interestingly, the A+F model appears to generally outperform other models with the average rank 1.89 and average z-score 0.606, followed by A+P model as the favorable second-best model with the average rank 2.20 and average z-score 0.2217. These kinds of graph including FunctionalGroup graph or Pharmacophore graph, provide extra meaningful features which may be difficult for the model to learn by itself. In contrast, the JunctionTree graph contains simple structure-based features which may be moderately captured during embedding process. Consequently, the integration of data-driven ensembled information offers advantages for model learning. In conclusion, based on statistical results, the recommended approach is to use a combination of atom-level graph and functional group-based graph, as it positively enhances model learning performance across various tasks.

**Comparison with other models.** The proposed model architectures are further performed additional experiments to demonstrate their competitive

**Table 3 | Summary of model performance ranking of 2-graph models for all datasets with all architectures**

| Model | MMGX | | GCN | | AttentiveFP | | AvgRank | AvgZScore |
|---|---|---|---|---|---|---|---|---|
| | AvgRank | AvgZScore | AvgRank | AvgZScore | AvgRank | AvgZScore | | |
| A | 3.20 | −0.6017 | 3.67 | −1.2060 | 2.87 | −0.4017 | 3.24 | −0.7364 |
| A+F | 2.13 | 0.5064 | 1.60 | 0.8176 | 1.93 | 0.4778 | 1.89 | 0.6006 |
| A+P | 1.93 | 0.3155 | 2.33 | 0.1643 | 2.33 | 0.1853 | 2.20 | 0.2217 |
| A+J | 2.73 | −0.2202 | 2.40 | 0.2240 | 2.87 | −0.2615 | 2.67 | −0.0859 |

Note: The underlined numbers are the best performance of each dataset.

performance compared to other state-of-the-art methods. Five recent state-of-the-art methods focusing on supervised learning models for molecular property an activity prediction with alternative or multiple graph representations are selected for comparison with proposed 2-graph combination models. All five methods include PharmHGT (pharmacophoric-constrained heterogeneous graph transformer)[20], HimGNN (hierarchical molecular graph neural networks)[24], ML-MPNN (multi-level message passing neural network)[23], FunQG (novel graph coarsening framework utilizing functional groups based on a quotient graph)[16], and RG-MPNN (pharmacophore-based reduced-graph message-passing neural network)[19]. The method description and results are listed in Supplementary Note 9. In summary, although our proposed model architecture does not consistently outperform other state-of-the-art methods in every task, it demonstrates comparable and competitive performance. This confirms the promising practicality and validity of our proposed model architectures for further analysis and model interpretation.

Even though different graph representations and model schemes demonstrate promising performance, they are inconsistent across each dataset. Some combinations can positively improve model learning by providing meaningful features. On the other hand, some combinations can negatively reduce the performance by introducing irrelevant features, bias, and complexity. Thus, selection of molecular graph combinations with additional feature engineering and model architecture design should be appropriately considered to get better performance for a specific task depending on the nature of the datasets.

**Interpretation results**

To validate the interpretation, different views of interpretation from attention weights are visualized and compared with chemical background knowledge. Each view offers advantages for different applications in drug discovery.

**Single prediction view**. Firstly, the comparison of the attention weights with ligand interaction map is analyzed. The first example is from CYP2C8 dataset. The troglitazone molecule has been reported as a potential inhibitor for the CYP2C8 target[42]. The interpretation results using average attention weights from 5-fold model compared with the interaction map of the complex PDB:2VN0 are shown in Fig. 3. All models can positively identify the thiazolidinedione fragment, which contains ketone oxygen that can form a hydrogen bond with target residues corresponding with the report in[42]. However, most models are not able to give high focus to the part of ether oxygen in the middle of the molecule, which can form a hydrogen bond with the residues using nearby water molecules reported in[42] as well. Despite that, the model interpretations can relatively capture the important parts consistent with the interaction region reported in the literature.

Another example is from BACE dataset. Umibecestat or CNP520 was discovered as a potent small molecule for BACE-1 inhibitor[43]. The interpretation using average attention weights from 5-fold model are compared with the interaction map from PDB:6EQM, as shown in Supplementary Fig. S4. Even if most models cannot obviously recognize the part of the oxazine nitrogen that forms interactions with target residues, model A, A+F and A+J still greatly emphasize on the portion of oxazine that contributes to the binding site, as well as putting high attention weight to oxazine oxygen

that accepts H-bonds from water molecules, as discussed in[44]. This indicates that important binding regions for ligand activity are well captured by the model interpretation providing understanding of ligand-target interaction.

These kinds of results can be utilized to understand the binding activity and binding region of ligands. In addition, they can be applied for molecular optimization or simplification task for generating better optimal ligands while maintaining the affinity property.

The next analysis is the quantitative interpretation performance using synthetic binding logics with known ground truths. To better compare interpretation performance of different models, the statistical measurement should be analyzed. However, it is difficult to quantitatively assess the interpretation performance from the real-world dataset because of the complexity and no defined ground truth. Therefore, the synthetic binding logics datasets are exploited as the ground truths are completely predefined. The task is to evaluate the interpretation performance of the model whether it correctly assigns high attention weights to the relevant substructure(s) corresponding to the binding logics or not. This task is similar to classification task of node importance using attention weights; therefore, the performances are measured in Attention AUROC (AttAUROC) and Attention Accuracy (AttACC) metrics. See Materials and Methods section for the definition and calculation of these two metrics. The results from 5-fold model only true positive prediction are summarized in Table 4. According to the results, model A+J performs best in average performance and ranking. This would be because JunctionTree graph basically focuses on simple structural features that are enough for these logics. However, Atom graph performs best in Logic14 dataset which seems to be little more complex. This maybe because it gives attention to only few features that are enough for specific task.

To examine the results, some examples are investigated to see how model gives attention to the molecules. All of the following results are obtained from the models with the best performance on validation set. The first example is from Logic7 (unbranched alkane AND carbonyl) dataset, as shown in Fig. 4. Although the ground truths contain multiple important substructures, model A gives high attention to only one substructure. In contrast, the combination graph models precisely capture multiple important substructures aligning with the corresponding ground truths. Additional example of 3MR (3-membered ring) with the same analysis are reported in Supplementary Fig. S5. Nevertheless, when analyzing Logic14 datasets which is a little complex, from Supplementary Fig. S6, model A demonstrates best in this dataset as it can capture important substructure that is enough for prediction correctly. While model A+J and A+P do not only assign attention weights to the important substructure but also other irrelevant features as well. In summary, the combination graph models can still give attention to the most important substructure, but they may also find other hidden yet unrelated features which must be taken into concern during analysis. To address this issue, some research suggests ways to enhance attention efficiency, including information theory analysis with additional model elements[45], efficient attention[46], norm-based analysis attention weights[47], or focused attention with transformer-based models[48]. It is important to note that this framework is applicable to other explainable AI techniques besides attention as well.

To elaborate the exploration of information interactions among the diverse views, additional analysis to examine the correlation between

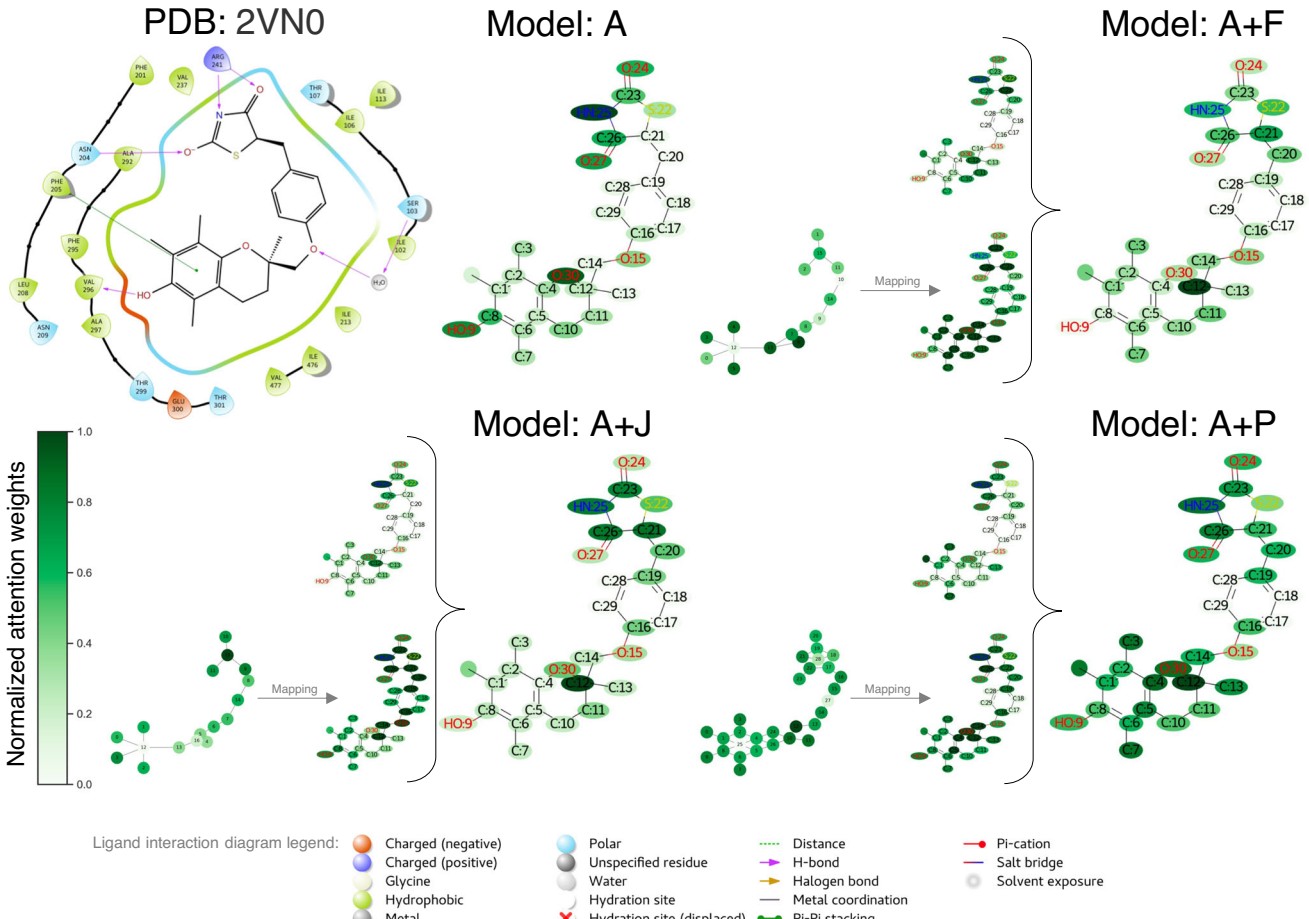

**Fig. 3 | Interpretation on single prediction view of troglitazone and the interaction map with CYP2C8 complex (PDB:2VN0).** All visualizations of model attention follow the interpretation extraction procedures. For the 2-graph scheme, the mapping and combining processes are applied to visualize on original atom-level graph. The ligand interaction diagram is calculated and generated by Maestro version 12.4.072, Schrödinger software suite with the legend below.

attention from multiple graph combination models whether the model pays attention to correlated regions of two graphs or not are conducted. The results are reported in Supplementary Fig. S7. Most of the models provide weak or even no correlation between two sets of attention from two graphs. These results indicate that models typically focus on different part of molecules. This situation can be considered beneficial, as several important regions in compound can be captured and emphasized via combining process when integrating multiple graphs.

**Node features view.** Analysis of significant node features using attention weights is examined. Taking AqSolDB dataset as an example, this dataset is a regression task predicting the aqueous solubility property of compounds. To demonstrate the important node features for a specific range of predictions, the compounds predicted as soluble and highly soluble (LogS > − 2) based on[49] are chosen for this analysis. The average attention weights and number of node feature for all 5-fold model are collected and plotted in the graph as shown in Fig. 5. The significant node features are observed in the area of high average attention weights and high number of feature nodes or on the top-right corner of the scatter plot. We can see that the node features with oxygen and nitrogen obtain more significance for soluble molecules. This is because they are likely to form hydrogen bonds with solvents. Interestingly, the carbon atom and the aromatic ring of carbon atoms mostly receive low attention weights for all models. Obviously, this perspective provides some useful trends to comprehend the datasets in the defined range. Visualization in node features view of combination graphs can clearly convey high-level information that is more meaningful than atom graph without further

processing, particularly model A+F and A+J as they provide node features in expressive functional groups and substructures. These results can be adopted for applications of knowledge extraction and trend analysis which would be useful for understanding particular property.

**Potential substructures view.** To understand the attention weights in more intuitive and comprehensive way, the interpretation in potential substructures view is introduced to suggest substructures highly influencing the tasks using pharmaceutical endpoint tasks datasets. Starting with potential substructures statistics, taking two classification datasets including AmesMutag and CYP2C8 as examples, the results are analyzed only the compounds predicted as positive class. After performing potential substructures extraction with the threshold given for each dataset, the number and instances of potential substructures for these two datasets that are significant for prediction of positive class from the model with the best performance in validation set are recorded in Fig. 6 and Supplementary Fig. S8. We can see that, different interesting potential substructures can be extracted from the models, and each substructure provides more comprehensive understanding with multiple atoms and functional groups.

Next, the above results are further compared with the reported key structural patterns from the literature to measure the capability of the model in capturing the potential substructures matching with those reported one. For AmesMutag datasets, many literatures studied about toxicity alerts[50–52]. Thus, the potential substructures from positive compounds should align with those toxicity alerts. As well as CYP2C8, some studies reviewed about significant fragments of ligands for this target[27,42], therefore; the interpretation

**Table 4 | Interpretation performance of synthetic binding logics datasets in AttAUROC and AttACC**

| Metrics | Model | Logic6↑ | Logic7↑ | Logic9↑ | Logic14↑ | 3MR↑ | Avg | AvgRank |
|---|---|---|---|---|---|---|---|---|
| AttAUROC | A | 0.8535 (0.2102) | 0.9777 (0.0482) | 0.8722 (0.1202) | <u>0.9975 (0.0015)</u> | <u>0.9964 (0.0017)</u> | 0.9395 | 2.20 |
| | A+F | 0.9888 (0.0097) | 0.8761 (0.0202) | <u>0.9656 (0.0256)</u> | 0.9293 (0.0202) | 0.9893 (0.0115) | 0.9498 | 2.60 |
| | A+P | 0.8869 (0.1545) | 0.8902 (0.0614) | 0.9477 (0.0479) | 0.8984 (0.0292) | 0.9554 (0.0745) | 0.9157 | 3.20 |
| | A+J | <u>0.9993 (0.0013)</u> | <u>0.9988 (0.0009)</u> | 0.8618 (0.1086) | 0.9949 (0.0029) | 0.9961 (0.0018) | <u>0.9702</u> | <u>2.00</u> |
| AttACC | A | 0.9084 (0.0008) | 0.8301 (0.0008) | 0.5824 (0.0015) | <u>0.9791 (0.0095)</u> | 0.9676 (0.0444) | 0.8535 | 2.80 |
| | A+F | 0.9636 (0.0234) | 0.8946 (0.0099) | 0.8610 (0.0007) | 0.8259 (0.1003) | 0.9515 (0.0387) | 0.8993 | 2.40 |
| | A+P | 0.8215 (0.1592) | 0.8612 (0.0086) | 0.7229 (0.1259) | 0.6348 (0.0565) | 0.8602 (0.1441) | 0.7801 | 3.60 |
| | A+J | <u>0.9853 (0.0232)</u> | <u>0.8985 (0.0033)</u> | <u>0.8731 (0.0187)</u> | 0.9633 (0.0302) | <u>0.9831 (0.0036)</u> | <u>0.9407</u> | <u>1.20</u> |

Note: AttAUROC (Attention AUROC) and AttACC (Attention Accuracy) calculation is described in the material and method section. The underlined numbers are the best performance of each dataset. The numbers in parentheses are the standard deviations.

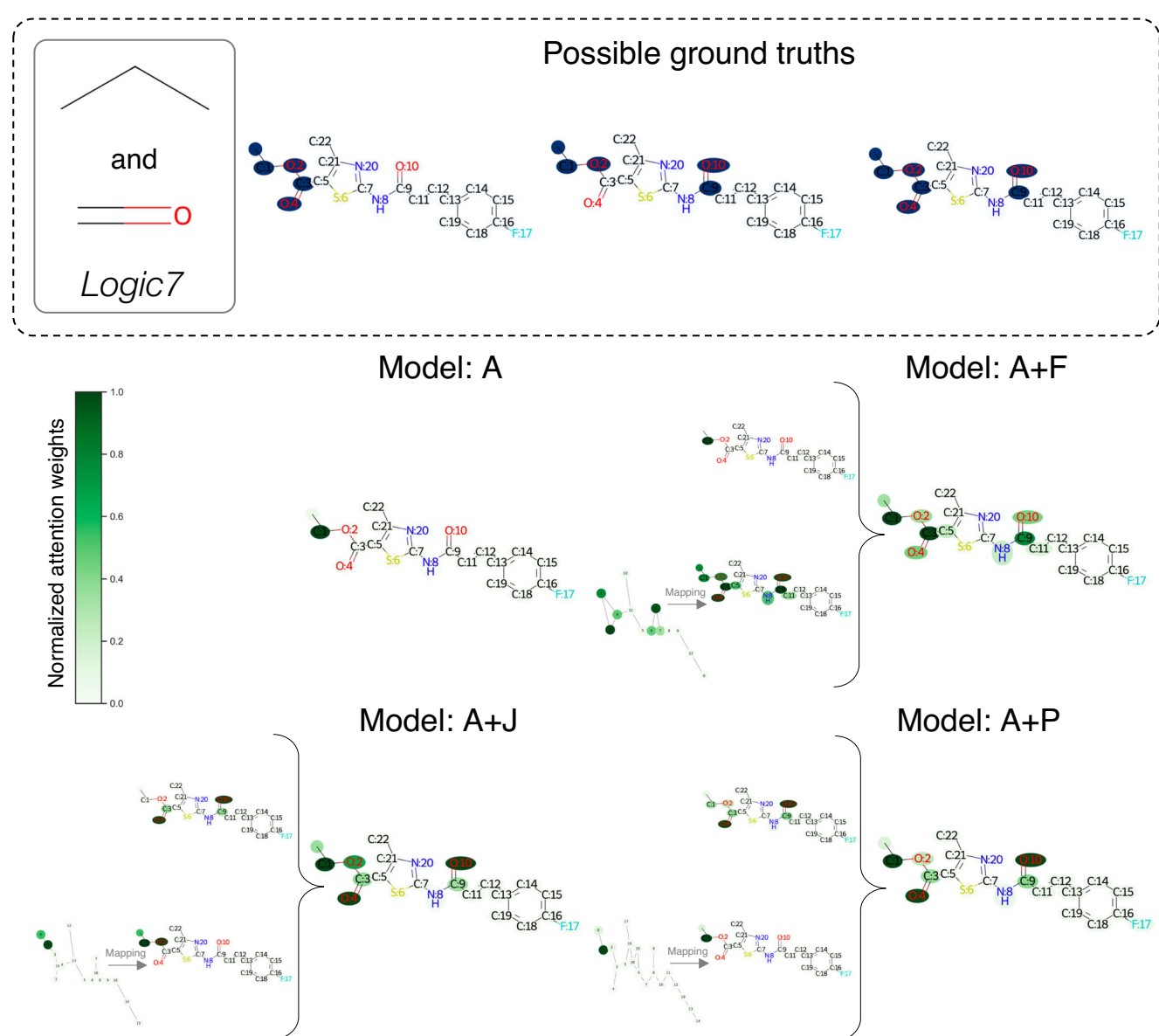

**Fig. 4 | Interpretation on single prediction view of sample molecule from Logic7 dataset with all possible ground truths.** All visualizations of model attention follow the interpretation extraction procedures. For the 2-graph scheme, the mapping and combining processes are applied to visualize on original atom-level graph.

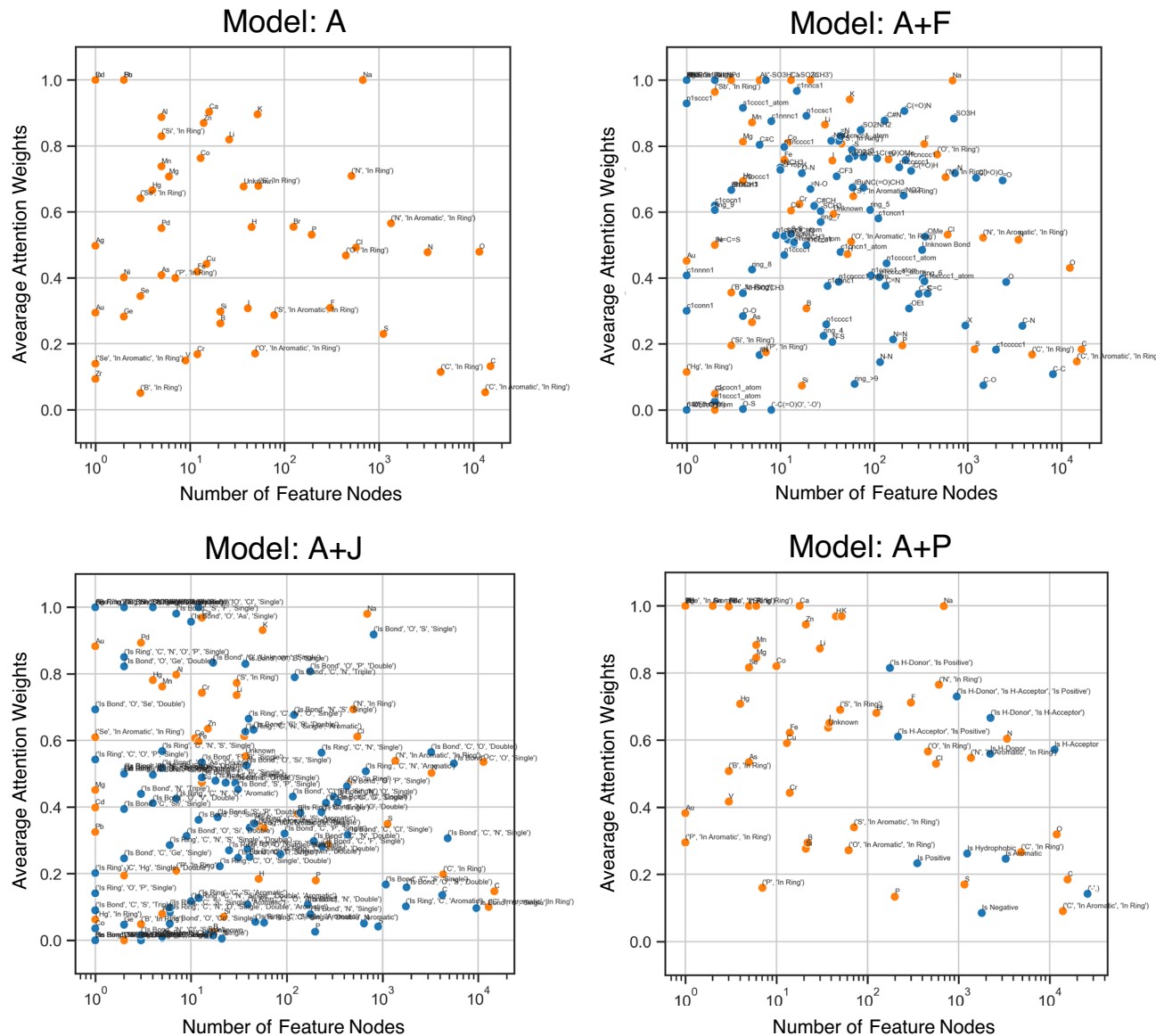

**Fig. 5 | Interpretation on node features view of AqSolDB dataset.** These graphs plot the average attention weights of each node feature with the number of feature nodes in the entire dataset. The orange dots represent the node features from the Atom graph. The blue dots represent the node features from the reduced molecular graph according to the scheme.

should also capture the relevant substructures that match with these fragments as well. As a result, most models provide acceptable matching outcomes for all datasets. According to Fig. 6 of AmesMutag dataset, fragments of nitro, nitroso, three-membered heterocycle, and chlorine can be captured by many models. Model A and A+J can recognize bromine (Br) atom (indicated by red circle) which is an important alert reported in[51], and model A, A+F, and A+P can also recognize sulfonate-bonded carbon atom groups (indicated by blue circle) which are also important as well. Another example is from CYP2C8 dataset as reported in Supplementary Fig. S8. While model A seems to perform best in this case, model A+F extracts larger potential substructures than other models. In summary for this part, the results are consistent with background knowledge and all potential substructures enhance more understanding and can be used for structural modification, task-specific molecule generation, and database creation. However, there are some shortages that the new knowledge of potential substructures is less captured as the extraction process is based on the frequency. Besides, the combination models sometimes return small number of potential substructures so extraction parameters should be studied and adjusted appropriately further.

## Conclusion

We introduced several molecular graph representations through graph reduction techniques aimed at generating higher-level molecular graph features. These representations were thoroughly evaluated on various datasets for predicting molecular properties and activities. Our findings demonstrated that different molecular graph representations offered varying levels of information, influencing model learning and interpretation. Combination of atom-level graph and other knowledge-based reduced graph such as functional group or pharmacophore graph appears to be the good promising combination across dataset categories and model architectures based on the studies. While integrating multiple molecular graphs into the model led to some moderate performance improvements, the extent of these enhancements varied across datasets. Therefore, it became crucial to thoughtfully select graph representations and conduct feature engineering when constructing prediction models. In addition to the aforementioned molecular graph representations, there are other intriguing and plausible graph representations that merit exploration in future research. These alternatives include 3D molecular graphs, fragment-based molecular graphs, and learned molecular graph representations, all of which contribute valuable chemical features.

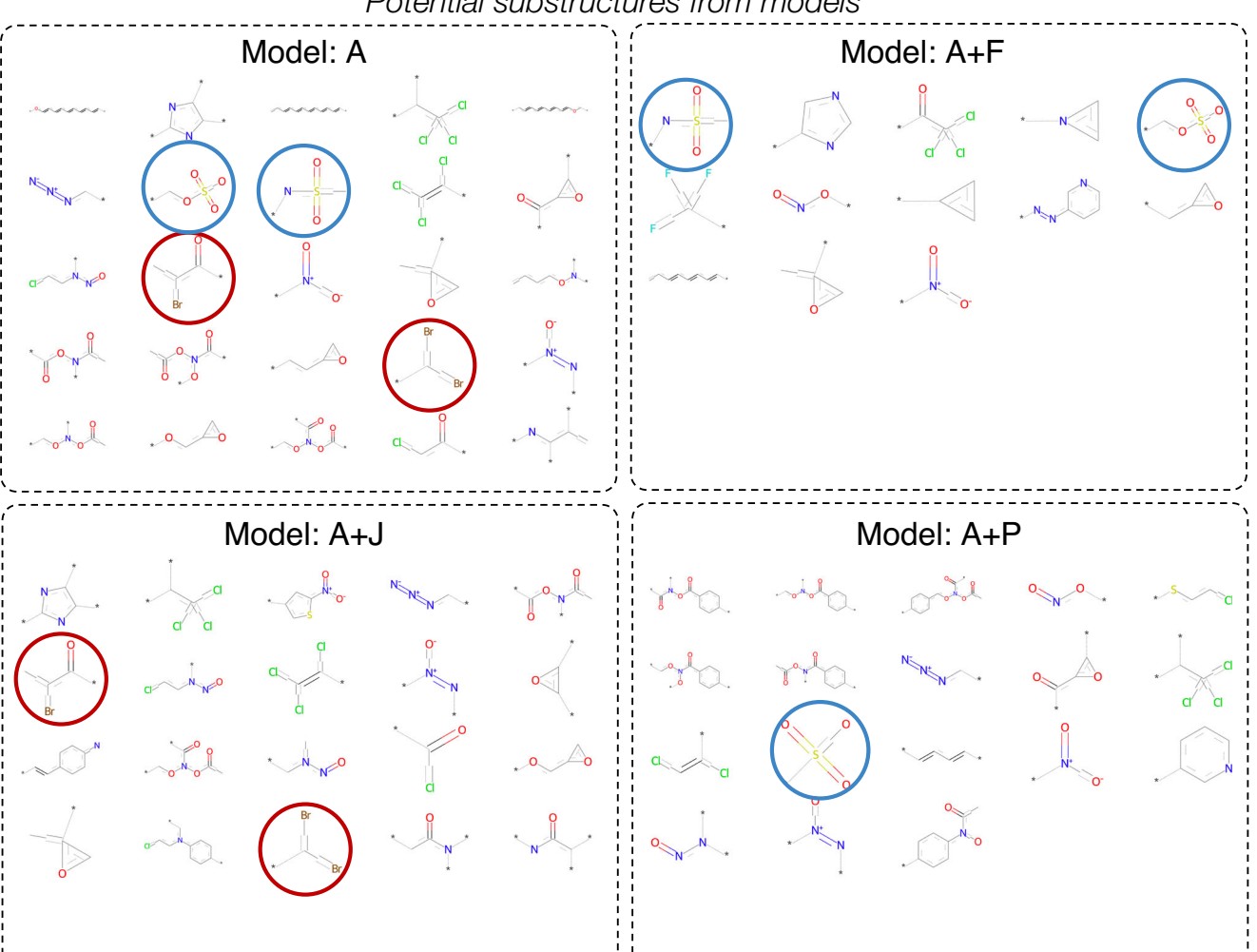

**Fig. 6 | Interpretation on potential substructures view of AmesMutag dataset.** Reported key structural patterns from the literature of AmesMutag dataset and potential substructures extracted from the models. The red circles indicates the bromine (Br) atom-related substructures and the blue circles indicates sulfonate-bonded carbon atom groups-related fragments.

Furthermore, integrating reduced molecular graph representation can greatly leverage the interpretation of the models by providing chemically meaningful node features for explanation aligning consistently with chemical knowledge. Atom with JunctionTree graph model performs the best for synthetic datasets tasks with quantitative evaluation in single prediction view. For node features view, combination models especially with FunctionalGroup or JunctionTree graph present more comprehensive features which are ready to understand. In potential substructure view, all models

provide reliable substructures consistent with background knowledge. On top of that, employing attention-based explanation techniques with multiple perspectives can enhance human comprehension of model predictions as well. All perspectives suggest different purposes and produce interesting findings and insights that could facilitate the subsequent processes, such as molecular simplification, structural modification, and lead optimization. The possible future direction would be the applications of interpretation results as the actionable outcomes to support other processes in drug discovery.

To summarize, the concept of utilizing multiple molecular graph representations and diverse interpretable views can be universally extended to various tasks, graph-based model architectures, and explanation techniques to enhance model learning and interpretation.

## Material and Methods
### Molecular graph representation implementation
The molecular construction, graph representation, and visualization are implemented by open-source package RDKit and NetworkX. Firstly, every molecule is represented as standard SMILES notation and converted to molecule object using RDKit function. Molecules with no bond, single-atom molecules, and molecules that produce any errors during conversion are not considered in this study. For a specific task, if there are the same molecules with different target values, the molecules are considered as conflict and will be removed.

Initially, the molecules are defined as undirected atom-level molecular graph $G(V, E)$ where $V$ is a set of atoms with index e.g. $v_1, v_2, ..., v_n$ where $n$ is number of atoms and $E$ is a set of bonds containing pairs of atoms with index e.g. $(v_i, v_j)$ where $i$ and $j$ are atom index. To perform graph conversion and reduction to different graph representation, the function $R^x$ is introduced to convert the atom-level molecular graph to another new graph, where $x$ is the type of molecule graph representation. Hence, $R^x(G) = G^x(V^x, E^x)$ where $G^x$ is a new graph representation in which $V^x$ is a set of nodes of substructure with index containing set of atoms with index e.g, node $v_1^x$ contains $(v_1, v_2, ..., v_n)$ atoms where $n$ is number of atoms in substructure $v_1^x$. $E^x$ is set of edges containing pair of nodes with index e.g. $(v_i^x, v_j^x)$ where $i$ and $j$ are node index. Node and edge attributes are represented as vector with different dimension depending on each type of representation.

The full lists of each molecular graph representation features are summarized in Supplementary Note 14.

### Model architecture
The model structure, inspired by AttentiveFP[36] and its variants[12,53], contains four main modules including node/edge encoding module, node embedding module with GNN, molecule embedding module with attention mechanism, and prediction module as shown in Supplementary Fig. S9. At first, node/edge encoding module encodes molecular graph node and edge features into initial fixed-size embedding using linear layer. Next, the node embedding module learns initial node embedding features by aggregation information from neighboring nodes using modified graph isomorphism network (GIN), that takes edge features in neighboring aggregation[54], integrated with gate recurrent units (GRUs) resulting in final node embedding. GIN seems to perform good at discriminative power and it has connection with Weisfeiler-Lehman isomorphism test[55]. GRUs learn controlling how much information to be aggregated or reserved during neighborhood aggregation[53]. Thirdly, the molecule embedding module utilizes the concept of virtual super node with virtual links that connect to all nodes in the molecular graph. This virtual super node learns to readout the embedding features from all nodes through virtual link using graph attention network (GAT) with GRUs as well resulting in final molecule embedding. At this step, the attention weights are assigned to each node in the graph during the readout process. These attention weights are assumed to be indicating how importance of each node for a particular prediction and will be interpreted as model explanation. Lastly, the prediction module combines all molecule embeddings from multiple graphs and executes classification or regression task to generate prediction results using fully connected layers.

### Experimental design
The extensive experiments are conducted on several datasets with multiple molecular graph representations. There are four main schemes for this study which are Atom graph (A), the combinations of the atom graph with another reduced molecular graph, which are Pharmacophore graph (A+P), FunctionalGroup graph (A+F), and JunctionTree graph (A+J). Apart from those schemes, additional experiments are conducted on reduced graph only (F, J, P), 3-graph (A+F+J, A+F+P, A+J+P), and 4-graph (A+F+P+J) models as well. In the case of multiple graph combination model using Atom graph and one or more reduced graphs, the initial pooling process will be performed to initiate reduced graph node features by integrating Atom graph node features to reduced graph node features. Apart from the original reduced graph node features, the reduced graph node features are extended with the pooled node features from the corresponding nodes in Atom graph using sum-pooling as shown in Supplementary Fig. S9. This process could help enriching the reduce graph node features so that the information of original atom-level graph is not completely abandoned. In multiple graph combination model, each graph is learned independently through the GNN backbone and attention mechanism. Subsequently, the molecule embeddings from all graphs are concatenated to generate the final ensemble molecule embedding before being fed into the prediction module.

### Implementation and hyperparameter tuning
The dataset management and model operation are manipulated by open-source PyTorch and DeepChem libraries. The dataset splitting method is suggested based on the original paper if specified; otherwise, random splitting is used. Typically, the datasets are split into train with validation and test sets with the ratio of 8:2 respectively. The train with validation sets are split and trained using 5-fold cross-validation. The hyperparameter tuning is performed using Optuna library[56]. The full lists of hyperparameter are recorded in Supplementary Table S14. The molecule embedding size is fixed to 256 dimensions. The learning rate, weight decay, dropout rate and batch normalization are set up appropriately according to each dataset. All models are trained for 300 epochs, but training is stopped earlier when the performance of the validation set is not improved for consecutive 30 epochs. The average of AUROC and RMSE among 5-fold are reported as performance for classification and regression tasks, respectively.

### Interpretation extraction procedures
The interpretation of the model is derived during the molecule embedding module with attention mechanism. The procedures of extracting attention weights are displayed in Supplementary Fig. S10. For each graph, the attention weights at the links connecting to a virtual super node are extracted and normalized with min-max algorithm, representing the node attention as shown in Supplementary Fig. S10A. In cases involving reduced molecular graphs, the results can be interpreted either using the its node features directly or through a mapping process to visualize on the original atom-level graph. When using the mapping process, the attention weights at the node features are redistributed to the corresponding original atom-level graph nodes as illustrated in Supplementary Fig. S10B. If multiple nodes of the reduced graph correspond to the same node in the atom-level graph, the summation of those nodes' attention weights is used to build up the node importance. Subsequently, all attention weights are normalized with min-max algorithm as well. For combination schemes, the attention weights of reduced graph must be mapped back to the original atom-level graph first. Then, all mapped attention weights of corresponding nodes are combined by choosing the maximum attention weights among all graphs to emphasize on the focused part of the molecule with greater priority. This attention combining process is visually depicted in Supplementary Fig. S10C. These procedures is consistently applied to all types of reduced graphs. The extracted attention weights are visualized on the different perspectives to provide intensive and alternative understanding of the model interpretation.

## Interpretation views

There are three different interpretation views introduced in this study to provide complete comprehensive insight with different perspectives. Each view is evaluated based on different methods in qualitative or quantitative ways. Moreover, each view offers various outcomes that are suitable for certain applications as well.

**Single prediction view.** Single prediction view visualizes the interpretation for a particular single prediction. The attention weights are analyzed using interpretation extraction process to get the final atom-level molecular graph interpretation. In this study, the intensity of the green color is used to represent the level of importance for that particular node/atom.

To qualitatively assess this view, the average interpretation results from 5-fold model are examined with background knowledge. For the ligand binding activity, it is assumed that the interpretation should focus on the relevant region of binding activity. Therefore, interpretation results are then compared with the interaction map of complex structures between ligand and target to observe how model learns to classify active ligand. For quantitative evaluation, the synthetic binding logics with known ground truth are employed followed the work from[57]. Given the synthetic datasets which are created using exact logical operations of substructures including AND, OR, and NOT, the interpretation performance can be statistically evaluated through the classification of important nodes using attention weights whether the model can correctly give dominant attention weights to the nodes matching up with the logics ground truths or not. The evaluation metrics are adapted from classification task which are Attention AUROC (AttAUROC) and Attention Accuracy (AttACC).

Attention AUROC (AttAUROC) is defined as the area under the receiver operating characteristic curve from the final node attention weights compared with the ground truth at varying threshold values.

Attention Accuracy (AttACC) is defined as the accuracy for the important nodes from the attention mechanism (high attention weights) compared with the ground truth labels. The formula is defined as

$$AttACC = \frac{1}{N} \sum_{i=1}^{N} I(y_i - \bar{y}_i) \qquad (1)$$

where $N$ is the number of nodes in a molecular graph and $y_i$ is the ground-truth label for particular node $i$. $\bar{y}_i$ is the predicted importance of that node in which if the attention weight is greater than or equal to 0.5, the node is considered as important and predicted as 1, and the indicator function $I(x)$ returns 1 if $x = 0$, otherwise 0 is returned.

If the molecules have more than one possible ground truths, the maximum of evaluation metrics are recorded. The average of interpretation performance from 5-fold model is used for comparison.

Single prediction view provides the specific important portions of a single predicted molecule which are useful for guiding compound optimization and simplification task.

**Node features view.** Node features view visualizes interpretation in collections of node features from the entire dataset. The average attention weights for each node features and the number of each node features are collected and plotted on the scatter plot. The significant node features can be recognized on the region of high average attention weights and the high number of feature nodes.

The evaluation can be analyzed with chemical background knowledge. For a particular range of prediction, it is assumed that the model should give high attention weights to the most relevant node features, and that node features should have large enough amount to support the relevance and reliability.

Node features view gives a benefit for understanding of the collection of predictions in dataset used in knowledge extraction and analyzing for trend analysis.

**Potential substructures view.** Potential substructures view analyzes the interpretation of structural patterns as potential substructures to provide higher-level chemical understanding. This view is done on the pharmaceutical endpoint tasks datasets as they have been studied and reported several key structural patterns. The molecules are broken down into smaller fragments. The fragments containing high attention weights and passed all detection rules are gathered and defined as potential substructures. The procedure to extract potential substructures consists of three steps. The first step is the identification of important fragments for a single molecule. The compounds are fragmented using various fragmentation techniques, including BRICS[58], RECAP[59], and GRINDER[27] to get all possible and diverse fragments with 3–20 atoms without breaking any rings. Then, the important fragments are detected using the rule that if the fragment has a median of attention weights greater than $P_f$ percentile of molecule attention weights, it is labeled as important fragments. The $P_f$ value is given based on dataset, by default 75 is used. The second step is the calculation of statistical values for each fragment. Three basic statistical values are the number of compounds having that fragment, the number of compounds having that fragment, which is labeled as important, and the important fragment percentage between those above two values. Next, the fragment importance score ($Score_{frag}$) from[60] is computed to quantify the significance of the fragments as shown in this formula.

$$Score_{frag} = \frac{\sum_{n=1}^{N_{frag}} \left( M_{frag(n)} - M_{mol(n)} \right)}{N_{frag}} \qquad (2)$$

Given a particular type of fragment, $n$ is an individual fragment, $M_{frag}$ is the average attention weights of fragment, $M_{mol}$ is the average attention weights of the molecule, and $N_{frag}$ is the number of fragments. The last step is the selection using conditions. The fragment is selected as a potential substructure if it meets all conditions which are: 1) The important fragment percentage is greater than or equal to 50%. 2) The number of compounds having that fragment which is labeled as important is large enough for each dataset. That number should be greater than or equal to $P_s$ percentile among all fragments. The $P_s$ value is depended on statistical results as well, by default 70 is used. 3) $Score_{frag}$ is greater than zero ensuring great significance. After that, the redundancies of substructures are removed, and the results are the final potential substructures.

To evaluate this, the chemical characteristics of potential substructures are analyzed and evaluated between each model with the reported key structural patterns. The study measures the performance of interpretation by observing the substructural matching between potential substructures from models and the key structural patterns reported from literatures.

Potential substructures view provides inspiration as a guideline for interesting and feasible structural modification, molecule generation, fragment collection creations, and ensuing optimization processes.

## Data availability

All datasets are publicly available in the original mentioned paper.

## Code availability

Codes are publicly available on GitHub https://github.com/ohuelab/MMGX.

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

## Acknowledgements

This work was financially supported by the Japan Science and Technology Agency FOREST (Grant No. JPMJFR216J), Japan Society for the Promotion of Science KAKENHI (Grant Nos. JP23H04880 and JP23H04887), and Japan Agency for Medical Research and Development Basis for Supporting Innovative Drug Discovery and Life Science Research (Grant No. JP23ama121026).

## Author contributions

M.O. conceived the study. A.K. designed and implemented the methods and computed results. M.O. and A.K. analyzed the results. A.K. drafted the manuscript and M.O. edited it. All the authors read and approved the manuscript.

## Competing interests

The authors declare no competing interests.
