## [Peer review file · Communications Chemistry]

Enhancing property and activity prediction and interpretation using multiple molecular graph representations with MMGXReviewers' comments:

Reviewer #1 (Remarks to the Author):

In this paper, the authors suggest that there has been a lot of recent research using simplified graphs to integrate alternative representations of advanced chemical information, but little research on the applicability and impact of these simplified molecular graphs on model learning and interpretation. Therefore, the author proposes MMGX (Multiple Molecular Graph eXplainable discovery). This paper explores the influence of multimolecular graphs including Atom, pharmacophore, JunctionTree and FunctionalGroup on model learning and interpretation from multiple perspectives. This study demonstrates that combining multiple graphs can improve model performance, but to varying degrees depending on the data set, mainly because multiple graphs can provide a more comprehensive substructure with consistent features and background knowledge. This manuscript is well written and useful for understanding model decision making, the applicability of multimolecular representation and the concept of different interpretive perspectives to tasks, architectures and interpretive techniques. While the issues involved in this research are very interesting, I have a series of issues that need to be addressed before the manuscript is suitable for publication. Main comments:

1. The interaction pattern between troglitazone and CYP2C8 complex in FIG. 4 does not indicate the structure and specific force of interaction, please add (same as in FIG. 5);
2. Please explain the process of converting different reduced diagrams into atomic-level diagrams for attention weight visualization (Figure 4,5,6,7,8 all have the same problem);
3. Please explain the process of combining attention weights of different molecular graphs in the combinatorial graph model (Figure 4,5,6,7,8 all have the same problem);
4. Please elaborate on the definitions and calculation formulas of the indicators AttAUROC and AttACC in Table 7,8;
5. The Logic7 dataset in the header of Figure 8 should be changed to the Logic14 dataset;
6. In FIG. 10, toxic related fragments such as bromine (Br) substructure and sulfonic acid bond carbon atomic groups can be marked with red circle symbols, which can help readers quickly and clearly understand the advantages of the combined graph model in extracting potential substructure;
7. Subsequent experiments can complement models of different reduced diagram combinations. Considering the possibility of information overlap and interference with atomic diagram combinations, the reduced diagram combination experiments can be expanded for further interpretative analysis.
8. Last but not least, new recently reported deep learning methods for molecular properties prediction tasks, such as Imagemol, MG-BERT, FP-GNN, K-BERT, FG-BERT, Mole-BERT, and HiGNN, should be reviewed and cited in the introduction section.

Reviewer #2 (Remarks to the Author):

This study delves into multiple molecular graph representations and their application in predicting molecular properties and activities. It explores a wide range of diverse graph views for molecules, while most previous studies, such as ReLMole (DOI: 10.1021/acs.jcim.2c00798) and HimGNN (DOI: 10.1093/bib/bbad305) which also proposed the utilization of multiple graph views, typically investigated two or three views. Additionally, the authors make an effort to derive interpretations from various levels of graph views. On the whole, the concept of employing multiple graph views is intriguing. However, there are several major concerns that need to be addressed.

1. I ponder upon the necessity of incorporating such a myriad of graph views, given the overlap in information conveyed by these perspectives. For instance, the Pharmacophore view and JunctionTree share a substantial portion of the graph, raising questions about redundancy.
2. The lack of comparison with the state-of-the-art methods for predicting molecular properties and activities presents a major issue in this study. It seems that the investigation primarily focuses on the impact of combining different views, while practical applications and utility of the proposed method

remain unclear. A lot of studies on molecular property and activity prediction have emerged in recent years (e.g. DOI: 10.1038/s41467-023-43214-1)

3.Regarding the model architecture, the employed backbone models are established Graph Neural Networks. However, there is a lack of explicit design explaining how information from different views is effectively integrated (a concatenation operation is used in this study). Furthermore, there is no exploration of information interactions among the diverse views, leaving room for potential improvements in capturing the synergies between them.

Reviewer #3 (Remarks to the Author):

This manuscript introduces MMGX (Multiple Molecular Graph eXplainable discovery) to investigate the effects of multiple molecular graph representation using several reduction techniques on model learning and interpretation with various perspectives. The evaluation results show that multiple graphs modestly improve model performance but in varying degrees depending on datasets. Interpretation based on multiple graphs provides a comprehensive analysis and attention-based attribution on molecular substructures seem to align well with domain knowledge in the selected examples.

Overall, I think this work is interesting but I hesitate to recommend its publication. Major concerns are summarized below.

Key Issues

#1. The idea of incorporating multiple views (or graphs in this case) to build a machine learning model is not new, there are various kinds of multi-view learning, sometimes also loosely termed multi-modality learning in Chemistry, that should be relatively straightforward to adapt to the multi-graph representations introduced here. Therefore, I do not see too much novelty/innovative aspect in this work.

#2. Furthermore, without comparing against other approaches, it is difficult to judge if the present approach signifies an appreciable advancement in the development of a molecular property predictor.

#3. It would be helpful to provide a concise summary of how to efficiently decide what could be the best multi-graph representation for any property prediction task.

#4 Authors acknowledged that the multi-graph based interpretation does not always give more accurate explanation. In fact, some graphs can incorrectly attribute to irrelevant molecular substructures. I think this is not a minor issue, and expect the authors to at least suggest some ways to mitigate this negative effect.

Authors' Reply

To the review of

Enhancing property and activity prediction and interpretation using multiple molecular graph representations with MMGX

Submitted to Communications Chemistry

Dear Reviewers,

We would like to express our gratitude to the reviewers of the manuscript for providing insightful feedback and comments. We found the reviewers' comments to be exceedingly helpful and have subsequently revised the manuscript accordingly. Below are the authors' response to the review of the manuscript.

Thank you for your time and kind consideration.

Sincerely yours,
Masahito Ohue

Replies to the reviewers' comments

Reviewer #1 (Remarks to the Author):

In this paper, the authors suggest that there has been a lot of recent research using simplified graphs to integrate alternative representations of advanced chemical information, but little research on the applicability and impact of these simplified molecular graphs on model learning and interpretation. Therefore, the author proposes MMGX (Multiple Molecular Graph eXplainable discovery). This paper explores the influence of multimolecular graphs including Atom, pharmacophore, JunctionTree and FunctionalGroup on model learning and interpretation from multiple perspectives. This study demonstrates that combining multiple graphs can improve model performance, but to varying degrees depending on the data set, mainly because multiple graphs can provide a more comprehensive substructure with consistent features and background knowledge. This manuscript is well written and useful for understanding model decision making, the applicability of multimolecular representation and the concept of different interpretive perspectives to tasks, architectures and interpretive techniques. While the issues involved in this research are very interesting, I have a series of issues that need to be addressed before the manuscript is suitable for publication. Main comments:

1. The interaction pattern between troglitazone and CYP2C8 complex in FIG. 4 does not indicate the structure and specific force of interaction, please add (same as in FIG. 5);

Authors' reply:

We appreciate the remark. We have altered the image of ligand interaction diagram calculated and generated from Maestro version 12.4.072, Schrödinger software suite with the legend in those two figures already. Also, we mentioned the source of image in the figure's caption as well.

2. Please explain the process of converting different reduced diagrams into atomic-level diagrams for attention weight visualization (Figure 4,5,6,7,8 all have the same problem);

Authors' reply:

Thank you for your comment. We have added more explanations about the process of converting reduced graph into atom-level graph for attention visualization together with the figure for better understanding in the section *4.5 Interpretation Extraction Procedures*. All figures containing attention visualization are captioned with reference to the interpretation extraction procedures.

3. Please explain the process of combining attention weights of different molecular graphs in the combinatorial graph model (Figure 4,5,6,7,8 all have the same problem);

Authors' reply:

Thank you for your comment. We also have added more explanations about the process of combining attention from multiple graphs for attention visualization together with the figure for better understanding in the section *4.5 Interpretation Extraction Procedures*. All figures containing attention visualization are captioned with reference to the interpretation extraction procedures as well.

4. Please elaborate on the definitions and calculation formulas of the indicators AttAUROC and AttACC in Table 7,8;

Authors' reply:

Thank you for your comment. We have added the explanation regarding the definition and calculation of Attention AUROC (AttAUROC) and Attention Accuracy (AttACC) on the Materials and Methods section in Section *4.6.1 Single Prediction View*. Basically, this measurement is designed based on the concept of classification prediction to classify the importance of nodes using attention weights compared to the ground truth. The measurement is adopted from the work [1]. We have also elaborated the text and notes of Table 7,8 to refer to specified section describing the definition and calculation of these metrics as well.

5. The Logic7 dataset in the header of Figure 8 should be changed to the Logic14 dataset;

Authors' reply:

We appreciate the remark. We have edited the figure 8's caption from to Logic14 already.

6. In FIG. 10, toxic related fragments such as bromine (Br) substructure and sulfonic acid bond carbon atomic groups can be marked with red circle symbols, which can help readers quickly and clearly understand the advantages of the combined graph model in extracting potential substructure;

Authors' reply:

Thank you for your suggestion. We have marked the related fragments mentioned in the manuscript with the circle symbols and referred them in the text as well.

7. Subsequent experiments can complement models of different reduced diagram combinations. Considering the possibility of information overlap and interference with atomic diagram combinations, the reduced diagram combination experiments can be expanded for further interpretative analysis.

Authors' reply:

Thank you for your interesting idea regarding subsequent analysis for model enhancement with different reduced molecular graphs. When designing the multiple graph combination models, we choose the Atom graph as the base, as it typically contains fundamental information about molecular structures. On the other hand, reduced graphs contain incomplete information but include high-level features not defined in the Atom graph. Therefore, integrating reduced graphs would complement both representations. Using only reduced graph combination would result in the abandonment of detailed important information and produce a deficient representation, leading to poorer performance, as demonstrated in the model learning performance shown in Table 6 (Model F, Model P, and Model J). Interpretation studies are reliable when the model performance is acceptably promising. However, future studies would be necessary for creating novel singular representations containing substantial necessary information for various tasks. Therefore, we have decided to leave this idea for future work.

8. Last but not least, new recently reported deep learning methods for molecular properties prediction tasks, such as Imagemol, MG-BERT, FP-GNN, K-BERT, FG-BERT, Mole-BERT, and HiGNN, should be reviewed and cited in the introduction section.

Authors' reply:

Thank you for your suggestions and for introducing various interesting research topics. We have revised the introduction section to incorporate additional background information about molecular property and activity prediction using several deep learning techniques, highlighting the utilizations of AI in drug discovery.

Reviewer #2 (Remarks to the Author):

This study delves into multiple molecular graph representations and their application in predicting molecular properties and activities. It explores a wide range of diverse graph views for molecules, while most previous studies, such as ReLMole (DOI: 10.1021/acs.jcim.2c00798) and HimGNN (DOI: 10.1093/bib/bbad305) which also proposed the utilization of multiple graph views, typically investigated two or three views. Additionally, the authors make an effort to derive interpretations from various levels of graph views. On the whole, the concept of employing multiple graph views is intriguing. However, there are several major concerns that need to be addressed.

1. I ponder upon the necessity of incorporating such a myriad of graph views, given the overlap in information conveyed by these perspectives. For instance, the Pharmacophore view and JunctionTree share a substantial portion of the graph, raising questions about redundancy.

Authors' reply:

Thank you for your concerns. We would like to elaborate the differences between the node and edge features of each graph to demonstrate the unique information provided by each. The table below showcases the node and edge feature sets for each graph. It is evident that each graph contains distinct sets of features due to differing transformation rules. For example, the Atom (A) graph provides node features for any atom type, whereas the FunctionalGroup (F) graph does not contain any atom type node information, but instead, it contains predefined bond types, ring types, and functional groups. In the case of the JunctionTree (J) graph, it encompasses compositions of bonds or rings indicated by the number of each atom symbol, rather than relying only on predefined substructures. On the other hand, the Pharmacophore (P) does not include atom symbol information but utilizes only six types of pharmacophoric features (H-donor, H-acceptor, positive, negative, hydrophobic, and aromatic) which are totally distinct from JunctionTree (J) and FunctionalGroup (F) graph. In term of edge features, all reduced graphs utilize their own types of node connections for edge information, which also differ as well. We have included this table in the supplementary information F for clarification purposes. More detailed information about each graph's features for implementation is listed in the supplementary information E. Therefore, we confirm that although all graphs are considered complementary views, they are designed to represent different information, which would be useful for incorporation into a multiple graph combination model and would definitely impact model learning and interpretation.

Table 2-1A Summary of Difference between Molecular Graph Representations

Graph	Node Features					Edge Features
	Atom	Bond	Ring	Functional group	Pharmacophore	Edge
Atom (A)	Any					Chemical Bond
Pharmacophore (P)			Aromatic only		6 Types	Connection type
JunctionTree (J)	Junction Atom	Any (composition)	Any (composition)			Connection type
FunctionalGroup (F)		Predefined bond types	Predefined ring types	Predefined functional groups		Connection type

2. The lack of comparison with the state-of-the-art methods for predicting molecular properties and activities presents a major issue in this study. It seems that the investigation primarily focuses on the impact of combining different views, while practical applications and utility of the proposed method remain unclear. A lot of studies on molecular property and activity prediction have emerged in recent years (e.g. DOI: 10.1038/s41467-023-43214-1)

Authors' reply:

Thank you for raising this concern. We have conducted additional experiments to demonstrate the competitive performance of the proposed model architecture compared to other state-of-the-art methods. Currently, our focus lies on supervised learning models for molecular property prediction, and all methods were selected based on the concept of alternative or multiple graph representations published in recent years.

Five methods are used for comparison including, PharmHGT [2], which utilizes pharmacophoric-constrained heterogeneous graphs employing BRICS fragmentation for generating a pharmacophore view connecting with atom view with different edge types; HimGNN [3], which proposes hierarchical graph representations consisting of atom- and motif-based graphs using a custom motif extraction process; ML-MPNN [4], which creates a multi-level message passing neural network using four full ranges of levels, including nodes, edges, junction-tree subgraphs, and the entire graph; FunQG [5], which constructs a new molecular graph representation using functional groups as building blocks with the theoretical concept of a quotient graph; and RG-MPNN [6], which demonstrates the integration of pharmacophore information hierarchically into atom-level graphs with a message-passing neural network architecture.

We reproduce these methods from the online available code repository indicated in the respective publications, using the same sets of dataset splittings of the general benchmark dataset from MoleculeNet used in our research. It should be noted that each research has a different preprocessing policy to handle molecule data and model architectures.

The results are summarized in Fig. 2-1A and 2-2A. Although our proposed model architecture does not consistently outperform other state-of-the-art methods in every task, it demonstrates comparable and competitive performance. With our straightforward model architecture, this confirms the practicality and validity of our proposed model architectures. We have included this analysis in the supplementary information and mentioned it in the main manuscript.

It is important to highlight that our research primarily focuses on the impact of different multiple graph combination models rather than solely aiming for achieving the best performance. We have emphasized and modified our objective to clarify that this research aims to improve understanding and the selection of multiple molecular graph representations, as well as to develop model explanation analysis through various views of interpretations. Therefore, the practical applications and utility of the proposed method can contribute to the rational design of other research efforts aiming to enhance their GNN model learning and interpretation. This research

showcases the broad applicability of different model architectures and explainable AI techniques as well.

Figure 2-1A: Comparison of the performance from the proposed model and state-of-the-art methods on regression task.

Figure 2-2A: Comparison of the performance from the proposed model and state-of-the-art methods on classification task.

3.Regarding the model architecture, the employed backbone models are established Graph Neural Networks. However, there is a lack of explicit design explaining how information from different views is effectively integrated (a concatenation operation is used in this study). Furthermore, there is no exploration of information interactions among the diverse views, leaving room for potential improvements in capturing the synergies between them.

Authors' reply:

Thank you for your valuable comments. The model architecture used in this study is designed to be the representative custom interpretable model containing fundamental modules of GNNs. In this study, the integration of information from different views can be explained in two parts, as outlined in Section 4.3 Experimental Design, and depicted in Fig. 12. Firstly, for multiple graph combination models, the model is designed to include one Atom graph and one or more reduced graph(s). When a reduced graph is employed, a special process called pooling is utilized to enrich the information of the reduced graph. Therefore, this method integrates the original reduced graph node features with Atom graph node features, enhancing the richness of information in the reduced graph node features. Another integration occurs before the prediction module, where the molecule embeddings from all graphs are concatenated together, as the reviewer mentioned. These design choices aim to provide a simple and general concept of multiple representation integration, allowing for the observation of the attention mechanism from both graphs independently and effectively. We have added more information about the concatenation process in the manuscript for clarification.

About exploration of information interactions among the diverse views, we have conducted additional analysis to examine the correlation between attention from multiple graph combination models whether the model pays attention to correlated regions of two graphs or not. The models with the best performance on validation set are selected for this study. Firstly, the combination models are tested with all datasets, and the attention weights at the molecule embedding step are extracted. For the reduced graph, the attention weights are redistributed to the original atom-level graph through a mapping process as explained in the section *4.5 Interpretation Extraction Procedures*. Next, two attribute weight sets from a pair of graphs are compared using Spearman rank-order correlation which is a nonparametric measure of the monotonicity of the relationship between two sets of observations. After that, all Spearman correlation coefficients from every pair of graphs are recorded, and the histograms are plotted. The results of attention correlation from example datasets are visualized as shown in Fig 3A. We can see that most of the models provide weak or even no correlation between two sets of attention from two graphs. These results indicate that models typically focus on different part of molecules. This situation can be considered beneficial, as several important regions in compound can be captured and emphasized via combining process when integrating multiple graphs. We put this analysis in supplementary information.

Fig 2-3A: Histogram of attention correlation from all molecules in dataset extracted from attention weights at molecule embedding module between Atom graph and reduced graph based on each schema. The correlation values are calculated using Spearman rank-order correlation coefficient. The lines are computed using kernel density estimate to smooth the distribution.

Reviewer #3 (Remarks to the Author):

This manuscript introduces MMGX (Multiple Molecular Graph eXplainable discovery) to investigate the effects of multiple molecular graph representation using several reduction techniques on model learning and interpretation with various perspectives. The evaluation results show that multiple graphs modestly improve model performance but in varying degrees depending on datasets. Interpretation based on multiple graphs provides a comprehensive analysis and attention-based attribution on molecular substructures seem to align well with domain knowledge in the selected examples.

Overall, I think this work is interesting but I hesitate to recommend its publication. Major concerns are summarized below.

Key Issues

#1. The idea of incorporating multiple views (or graphs in this case) to build a machine learning model is not new, there are various kinds of multi-view learning, sometimes also loosely termed multi-modality learning in Chemistry, that should be relatively straightforward to adapt to the multi-graph representations introduced here. Therefore, I do not see too much novelty/innovative aspect in this work.

Authors' reply:

Thank you for your valuable opinion, and we apologize for ambiguity. Integrating multiple graphs for model learning has gained attention among researchers in recent years. We share this interest and aim to enhance understanding in this area. As outlined in the introduction, various molecular graph representation techniques have been applied in research, each potentially containing distinct information that could influence both model learning and interpretation. However, there remains an unexplored gap in the literature concerning the study and comparison of different multiple graph combination models, hindering progress in model design and understanding. Our study seeks to address this by investigating the effect of various molecular graph representations on model learning. Additionally, while some graph-based models and multiple-graph combination models integrate interpretation modules in their design, there has been limited analysis on how different molecular graph representations affect model learning. Furthermore, recent research has focused on a certain view of interpretation, potentially limiting the overall learning and scientific insight gained from the model. Hence, we propose to conduct analysis of interpretation from various perspectives to gain deeper insights. By introducing these new aspects including the exploration of different multiple molecular graph representations and diverse interpretation views, our study aims to contribute important benefits to researchers in this field, paving the way for enhancing model design and interpretation methodologies in molecular property and activity prediction.

#2. Furthermore, without comparing against other approaches, it is difficult to judge if the present approach signifies an appreciable advancement in the development of a molecular property predictor.

Authors' reply:

Thank you for your comment. We apologize for lacking this information. This question resembles question 2 from reviewer 2, so we would like to restate the answer here for further clarification. We have conducted additional experiments using five recent state-of-the-art methods related to supervised learning of alternative or multiple molecular graph representations for ligand property and activity prediction.

Five methods are used for comparison including, PharmHGT [2], which utilizes pharmacophoric-constrained heterogeneous graphs employing BRICS fragmentation for generating a pharmacophore view connecting with atom view with different edge types; HimGNN [3], which proposes hierarchical graph representations consisting of atom- and motif-based graphs using a custom motif extraction process; ML-MPNN [4], which creates a multi-level message passing neural network using four full ranges of levels, including nodes, edges, junction-tree subgraphs, and the entire graph; FunQG [5], which constructs a new molecular graph representation using functional groups as building blocks with the theoretical concept of a quotient graph; and RG-MPNN [6], which demonstrates the integration of pharmacophore information hierarchically into atom-level graphs with a message-passing neural network architecture.

We reproduce these methods from the online available code repository indicated in the respective publications, using the same sets of dataset splittings of the general benchmark dataset from MoleculeNet used in our research. It should be noted that each research has a different preprocessing policy to handle molecule data and model architectures.

The results are summarized in Fig. 3-1A and 3-2A. Although our proposed model architecture does not consistently outperform other state-of-the-art methods in every task, it demonstrates comparable and competitive performance. With our straightforward model architecture, this confirms the practicality and validity of our proposed model architectures. We have included this analysis in the supplementary information and mentioned it in the main manuscript.

We also believe that this research would be the appreciable advancement in the development of a molecular property predictor with the aim to provide understanding and representation design for improving model learning and interpretation.

Figure 3-1A: Comparison of the performance from the proposed model and state-of-the-art methods on regression task.

Figure 3-2A: Comparison of the performance from the proposed model and state-of-the-art methods on classification task.

#3. It would be helpful to provide a concise summary of how to efficiently decide what could be the best multi-graph representation for any property prediction task.

Authors' reply:

Thank you for your suggestions. We have adjusted the results and discussion sections, as well as the conclusion section, to provide a more concise summary of our findings. In terms of model learning performance, we conclude that the Atom with FunctionalGroup graph performs the best in terms of average rank and z-score, followed by the Atom with Pharmacophore graph. This indicates that integrating combination graphs with useful knowledge-based features improves model learning. Regarding interpretation, the Atom with JunctionTree graph model shows the best average rank for explaining synthetic datasets in the single prediction view, thanks to its ability to capture structural patterns. In the node features view, combination models, particularly those with the FunctionalGroup graph, present more chemically

intuitive results with apparent telling features. In the potential substructures view, all models provide reliable outcomes matching reported key structural patterns. We hope these conclusions help improve understanding of multiple molecule graph combination models and selection of graph representations for ligand property and activity prediction and contribute to the modeling design of future research.

#4 Authors acknowledged that the multi-graph based interpretation does not always give more accurate explanation. In fact, some graphs can incorrectly attribute to irrelevant molecular substructures. I think this is not a minor issue, and expect the authors to at least suggest some ways to mitigate this negative effect.

Authors' reply:

Thank you for your concern and advice. I agree with the issues that the reviewer pointed out. Attention mechanism can produce negative effect as demonstrated in the examples. Therefore, to address that issue, we have reviewed and added some suggestions regarding the ways to enhance the effectiveness of attention weights, including information theory analysis with additional model elements [7], efficient attention [8], norm-based analysis attention weights [9], or focused attention with transformer-based models [10]. We believe that attention is introduced to improved model learning and learned during the model training, thus, it is useful to be used as explanation. Besides attention, we would like to note that our framework and analysis is applicable for other explainable AI techniques as well.

We wish to thank the reviewers again for their valuable comments.

References

- [1] McCloskey, K., Taly, A., Monti, F., Colwell, L.J.: Using Attribution to Decode Binding Mechanism in Neural Network Models for Chemistry. *Proc. Natl. Acad. Sci. U.S.A.* 116, 11624–11629 (2019)
- [2] Jiang, Y., Jin, S., Jin, X., Xiao, X., Wu, W., Liu, X., Zhang, Q., Zeng, X., Yang, G., Niu, Z.: Pharmacophoric-constrained heterogeneous graph transformer model for molecular property prediction. *Commun. Chem.* 6, 60 (2023)
- [3] Han, S., Fu, H., Wu, Y., Zhao, G., Song, Z., Huang, F., Zhang, F., Liu, S., Zhang W.: HimGNN: a novel hierarchical molecular graph representation learning framework for property prediction. *Brief. Bioinformatics.* 24(5), bbad305 (2023)
- [4] Wang, Z., Liu, M., Luo, Y., Xu, Z., Xie, Y., Wang, L., Cai, L., Qi, Q., Yuan, Z., Yang, T., Ji, S.: Advanced graph and sequence neural networks for molecular property prediction and drug discovery. *Bioinformatics.* 38, 2579-2586 (2022)
- [5] Hajiabolhassan, H., Taheri, Z., Hojatnia, A., Yeganeh, Y.T.: FunQG: Molecular Representation Learning via Quotient Graphs. *J. Chem. Inf. Model.* 63(11), 3275-3287 (2023)
- [6] Kong, Y., Zhao, X., Liu, R., Yang, Z., Yin, H., Zhao, B., Wang, J., Qin, B., Yan, A.: Integrating Concept of Pharmacophore with Graph Neural Networks for Chemical Property Prediction and Interpretation. *J. Cheminform.* 14(52) (2022)
- [7] Wen, B., Subbalakshmi, K.P., Yang, F.: Revisiting Attention Weights as Explanations from an Information Theoretic Perspective. Preprint at <https://arxiv.org/abs/2211.07714> (2022)
- [8] Zhuoran, S., Mingyuan, Z., Haiyu, Z., Shuai, Y., Hongsheng, L.: Efficient Attention: Attention with Linear Complexities. In: 2021 IEEE Winter Conference on Applications of Computer Vision (WACV), Waikoloa, HI, USA, pp. 3530–3538 (2021)
- [9] Kobayashi, G., Kuribayashi, T., Yokoi, S., Inui, K.: Attention is Not Only a Weight: Analyzing Transformers with Vector Norms. In: Proceedings of the 2020 Conference on Empirical Methods in Natural Language Processing (EMNLP), pp. 7057–7075 (2020)
- [10] Gao, J., Shen, Z., Xie, Y., Lu, J., Lu, Y., Chen, S., Bian, Q., Guo, Y., Shen, L., Wu, J., Zhou, B., Hou, T., He, Q., Che, J., Dong, X.: TransFoxMol: predicting molecular property with focused attention. *Brief. Bioinformatics.* 24, 306 (2023)

Modification Logs

Section	Subsection	Modification
1 Introduction	-	 Added literature reviews on several deep learning techniques for molecular property and activity predictions Added more literature reviews about multiple molecular graph combination models Refined current limitations for both model learning and interpretation Revised and clarified research objectives
2. Results and Discussion	2.2 Molecular Graph Representations (2.2.1 Introduction to Molecular Graph)	 Emphasized on difference between molecular graph representations Referred to supplementary information about summary of difference between molecular graph representations
	2.3 Model Performance	 Added introduction Reorganized the model performance into subsections including:  Model Performance of 2-graph Combination Models Model Performance of Multiple-graph Combination Models Model Performance of Different Model Architectures Comparison with Other Models
	2.3 Model Performance (2.3.1 Model Performance of 2-graph Combination Models)	 Included overall results and summarization Added Table 6 summary of model performance ranking of 2-graph models for all datasets with MMGX Edited model performance of A+P on HIV1 dataset Edited average ranking of A and A+P in biophysics category Edited all average z-score in biophysics category
	2.3 Model Performance (2.3.3 Model Performance of Different Model Architectures)	 Included overall results from the experiments of 2-graph combination model with GCN and AttentiveFP model for all datasets Included summarization Added Table 8 summary of model performance ranking of 2-graph models for all datasets with all architectures
	2.3 Model Performance (2.3.4 Comparison with Other Models)	 Added new subsection Added model performance results from other state-of-the-art models Included summarization
	2.4 Interpretation Results (2.4.1 Single Prediction View - Comparing the attention weights with ligand interaction map)	 Changed image of ligand interaction diagram of PDB:2VN0 and PDB:6EQM of Fig. 4 And Fig. 5 using diagrams from Maestro, Schrodinger Edited above figure captions to described the attention mapping and combining process and ligand interaction diagrams Fixed typos CYP2C8
	2.4 Interpretation Results (2.4.1 Single Prediction View - Quantitative interpretation performance)	 Referred to the definition and calculation of quantitative interpretation evaluation metrics, AttAUROC and AttACC.

Section	Subsection	Modification
	using synthetic binding logics with known ground truths)	 • Provided suggestions on enhancement of attention mechanism for model interpretation • Edited figures' notes to refer to the definition and calculation of AttAUROC and AttACC metrics • Edited Fig. 6,7,8 captions to described the attention mapping and combining process
	2.4 Interpretation Results (2.4.3 Potential Substructures View)	 • Indicated mentioned substructures with colored circle symbols. • Edited Fig. 10 by indicating mentioned substructures with colored circle symbols.
3 Conclusion	-	 • Provided concise summary for good promising combination for model learning and advantages of each combination for model interpretation
4 Material and Methods	4.3 Experimental Design	 • Explained more about how to integrate multiple graph in combination model
	4.5 Interpretation Extraction Procedures	 • Explained more about how to extract, map, and combine attention weight in combination models • Added figure describing interpretation extraction procedures.
	4.6 Interpretation Views (4.6.1 Single Prediction View)	 • Added definition and calculation of metrics AttAUROC and AttACC
	4.6 Interpretation Views (4.6.3 Potential Substructures View)	 • Fixed fragment important score formula
References	-	 • Updated accordingly
Supplementary information	-	 • Updated accordingly

REVIEWERS' COMMENTS:

Reviewer #1 (Remarks to the Author):

All concerns have been resolved, recommended for publication.

Reviewer #2 (Remarks to the Author):

My concerns have been addressed.

Reviewer #3 (Remarks to the Author):

In this revised version of the manuscript, authors have improved their manuscript accordingly to the reviewers comments. I am happy to recommend the article for publication.